# Allocation of the household food budget among shopping basket items: How is it influenced by promotions?

**Wafa Mehaba**[1,2]*, **Djamel Rahmani**[1,2], **José Maria Gil**[1,2]

**1** Department of Agri-Food Engineering and Biotechnology, Polytechnic University of Catalonia, Castelldefels, Spain, **2** Center for Agrofood Economics and Development (CREDA-UPC-IRTA), Castelldefels, Spain

* wafa.mehaba@upc.edu

## Abstract

Retailers have been using promotion as a differentiation strategy that influences consumers' expenditures and their shopping basket budgetary allocation. This study assessed the effect of retail promotions on total shopping basket expenditure and determined whether promotions provoke a reallocation of the shopping budget. The analysis was performed on a chain of supermarkets in Catalonia, Spain using a consumer scanner data set from Kantar Worldpanel for 2017. The methodological approach had two steps: prediction of the effect of promotion on household expenditures using an expenditure regression model and estimation of the promotion own- and cross-effect using the censored Exact Affine Stone Index. Promotion had a positive own-effect and mostly a negative asymmetric cross-effect, implying a small but significant budget reallocation.

## 1. Introduction

Due to the monopolistic competition in the retail market in Spain over the last two decades, marketers have been striving to create competitive advantages to maximise their profits. This has led them to embrace various strategies and techniques to make their products stand out and to attract more customers. Among the strategies applied, differentiation is cited as the most attractive, and it can be quite heterogeneous between industries. It can be achieved by focusing on the product itself, the delivery system, the marketing strategy [1], offering innovative features, providing superior services, creating a solid brand name, implementing successful promotions and so on [2]. As a result, differentiation has never been more of interest than it is now. [3] stated that among the four Ps of marketing, promotion is taking an important position. [4] defined *promotion* by as 'an action-focused marketing event whose purpose is to have a direct impact on the behaviour of the firm's customer'. [5] defined *promotions* as 'marketing events limited in duration, implemented to directly influence the purchasing actions of customers, with the underlying intention of achieving the objectives set out in the marketing strategy for the retailer and/or manufacturer'. Hence, retailers allocate an important share of their marketing budgets to promotions [6–8]. Indeed, the advertising and promotional

**Data Availability Statement:** Data cannot be shared publicly because of confidentiality agreements. Data are available from the Ethic committee of the Centre for Agrofood Economics and Development (CREDA) (contact via zein.

kallas@upc.edu) for researchers who meet the criteria for access to confidential data. The adapted data underlying the results presented in the study are available from Centre for Agrofood Economics and Development (CREDA), creda@creda.es, https://www.creda.es/es/bienvenidos/.

**Funding:** This work was supported by the funding received from the AEI (Spanish Research Agency) under grant agreement No PID2019-111716RB-I00 (project acronym SUSPROMO).The funders had no role in study design, data collection and analysis, decision to publish, or preparation of the manuscript.

**Competing interests:** The authors have declared that no competing interests exist.

expenditures in Spain peaked in 2015–2019 (Fig 1), ranging from €2,290 in 2013 to €2,246.70 million in 2019 [9].

This important investment in promotion is mainly due to the stimulation that it can provoke in consumers' minds, pushing them to purchase more products or to purchase them faster or more frequently [11,12], increase their store visits [12,13] and buy new products or brands [14]. According to the report *Promotions, the Key to Winning Customers* prepared by the consultancy firm Kantar Worldpanel, [15], each promotional action brings an average of 1,113 new buyers to fast-moving consumer goods brands which means that 41% of the households that bought a promoted product in 2016 were not regular buyers of the brand. [16] classified the aforementioned effects into short-term effects, which appear at the time of the promotion, and long-term effects, which involve behaviour that takes place after the promotion. Considering the short term, various studies have focused on the influence of various promotions on consumer perception and purchasing behaviour, and on company profitability and performance. [17,18] maintained that promotions induce an increase in sales of the promoted products. [19] pointed out that aside from promotions' positive own effect, they can push consumers to buy non-promoted products. Other studies claimed that sales promotions can induce brand switching [8,20–22] or generate a cannibalisation effect [23]. Some other studies have focused on which promotional tools are most effective. For instance, [24] found that price discounts and more free products could have the greatest impact on consumer purchase behaviour. Moreover, [25] demonstrated that discounts induce purchase acceleration, stockpiling and product trials.

The aforementioned studies used different approaches to analyse promotion effectiveness. One of these approaches is the use of an elasticity-based technique [26]. The elasticity of category prices and promotion helps retailers to make crucial pricing decisions, such as choosing which categories are resilient to price increases and which should have a sales promotion [27]. For this purpose, [28] evaluated the effect of prices and advertising activities on sales of different types of milk and found that the response to promotions was positive and significant, varying from 0.167 for whole milk to 0.438 for 2% milk. [29] decomposed the total price elasticity of 173 brands across 13 product categories and noted that increased category purchasing (i.e., purchase acceleration) accounted for 25% of sales generated by price promotions. In the same context of measuring promotional effectiveness, [30] analysed category-level expenditures for

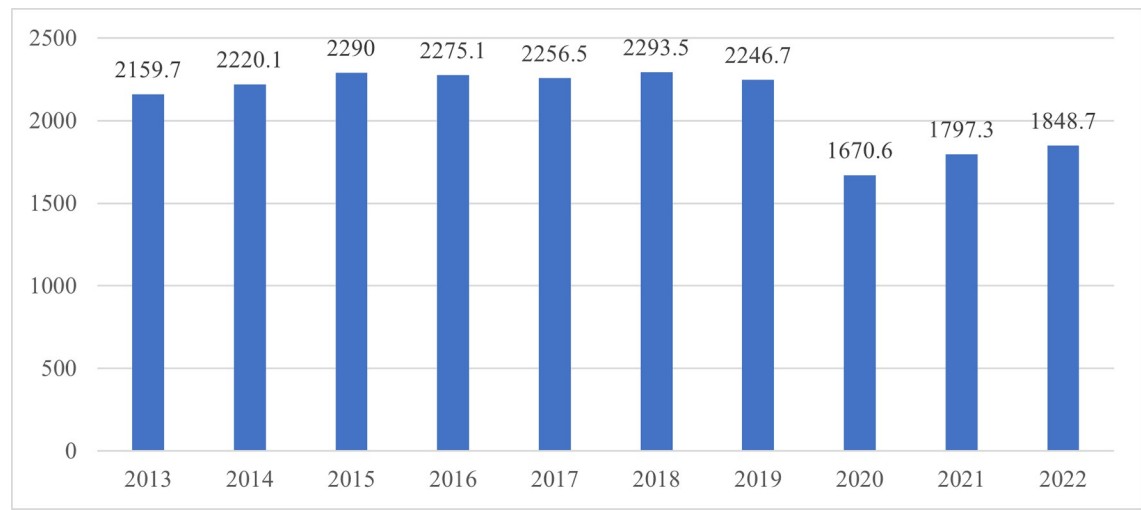

**Fig 1. Advertising expenditures in Spain, 2013–2022 (in million €) [10].**

labelled products and found that promotions had a relatively small but significant positive effect on category expenditures, even though such effect depended on the category, market and type of promotion. Similarly, [31], after analysing 560 product categories over four years, reported that in the short term, price promotions had an average elasticity of 2.21 and that more frequent price promotions increased such elasticity. Similarly, [32] examined the effects of price promotions on a low-priced food product category and found a 450% increase in the sales volume of the promoted brand inside the category and a 140% short-term increase in the category sales volume. Nevertheless, these studies focused on a specific brand or category, but the effect of promotions could be extended asymmetrically to product categories that are not affected by the promotion due to the income effects [33]. Cross-category effects suppose that a sales promotion for a product or a category could reduce sales of another product or category (illustrating the substitution effect) or raise them (showing the complementarity effect), which is what retailers desire. Consequently, retailers should have insights into interdependent product relationships to facilitate the allocation of the promotion budget [34]. For this purpose, [35] created a multicategory choice model that considers four categories. They noted positive own-promotion effects and lower cross-promotion effects; and the latter effects were asymmetric across food categories. Analogously, [34] utilised a model for two food categories (pasta sauces and pasta) to quantify the effects of cross-category promotion on purchase incidence, quantity decisions and brand choice. They found that disregarding these effects may cause retailers to have an inaccurate budget allocation and inaccurate promotion timing decisions. [36] measured the sales impact of temporary retail discounts on all brands within a product category and across subcategories and found that promotional price cuts offered on an item had both positive and negative effects on sales of competing items in the general product category. Similarly, [37,38] determined the own- and cross-promotional effects at the category level and discovered that a price promotion induces an increase in the category's revenue. In relation to the cross-category effect, both positive and negative effects were observed, although complementarity was the most dominant.

In these studies, different multivariate models were used to determine the effects of promotion on brand or category choice and the possible cross-effects on a limited number of categories. However, other researchers studied cross-price and cross-promotion effects using a demand system approach, consequently analysing a broader number of related categories. To study the impact of price and promotions (i.e., price discounts, featuring and display) on the demand for fish and seafood products, [39] used the Almost Ideal Demand System (AIDS) and reported that promotional impact is heterogeneous between categories, having both positive and negative effects. On the same topic, [40] reported a positive response to promotion for all types of seafood products and a heterogeneous cross-promotion effect that fluctuated from positive to negative, depending on the seafood category. More recently, [41] estimated an Exact Affine Stone Index (EASI) demand system to study the effect of promotions on different types of beverages and the banning of promotions of soft drinks in Scotland. They found a positive effect of promotions on different types of beverages, with substitution and complementarity cross effects.

The main shortcoming of the cited studies is that they restricted their analysis to a specific category of food products without considering the potential effects on the full shopping basket; that is, they assumed weak separability of the selected categories from the total food basket. These interdependencies are important because they help retailers to allocate their promotional budget across categories. Therefore, studying the full basket is a crucial component of the creation of any promotional mix in which retailers build new product bundles, provide special discounts, or put products in the best possible locations. To the best of our knowledge, very few studies have considered the effects of prices and promotions on the entire shopping

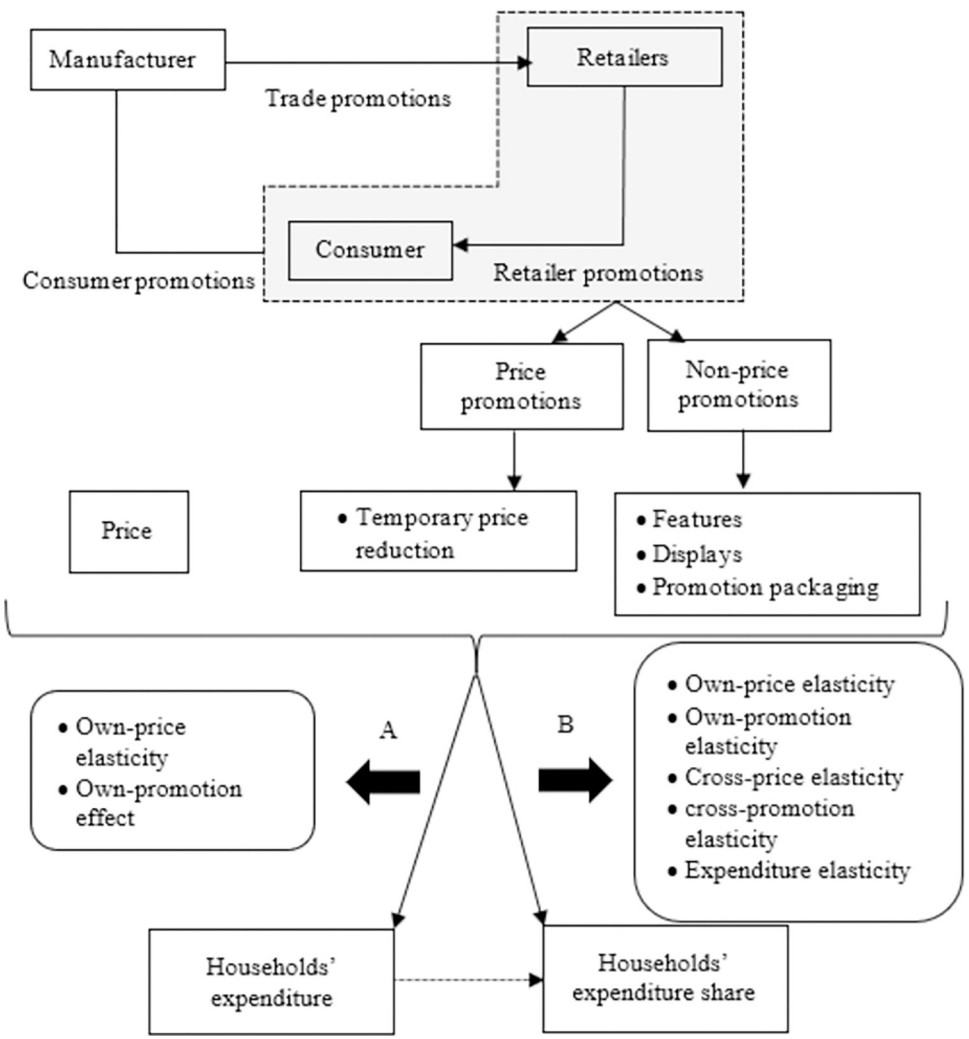

**Fig 2. Conceptual framework of this study.**

basket. [42] used panel data on household purchases from four stores of the same retailer, and [43] analysed the overall effect of promotions on consumers' food purchases in Scotland by area of deprivation to formulate food and health policies. In this study, we propose a conceptual framework (Fig 2) of the effect of sales promotions and prices on household expenditures to achieve our research objective: to determine the effect of sales promotions on households' allocation of their budget among the different products in a typical shopping basket in Spain. Our framework has two dimensions: (A) the direct effects of price and promotions on household expenditures, and (B) the possible cross-effects of price and promotions (i.e., substitution or complementarity), which could lead households to reallocate their budget.

Ultimately, we aim to contribute to the literature empirically and methodologically. Empirically, we aim to study household budget allocation decisions in Spain because retailers should be aware of how consumer responses to sales promotions vary between nations, states and provinces, as sales promotions are generally acknowledged as challenging to standardise due to the varying legal, economic and cultural factors that influence households and consumer behaviour. Consequently, we seek to provide retailers insights into the categories that are influenced most by promotions and how households manage their shopping budgets.

Methodologically, we use the EASI demand system because it allows for flexibility of Engel curve shapes, unlike the AIDS demand model. [44] demonstrated, using nonparametric regressions, that Engel curves can have different shapes. Additionally, budget share error terms can be interpreted as unobserved preference heterogeneity, as permitted by the implicit Marshallian demand system developed by [45] that merges the proprieties of the Hicksian and Marshallian demand functions. In addition, we introduce promotional indices into the EASI demand system that allow for the estimation of the own-promotion and cross-promotion elasticities for a broader number of food categories.

Therefore, we will strive to answer the following three major research questions:

- To what extent do promotion and price induce households to change their shopping expenditures?

- Which product category has the strongest effect on expenditures when sold under a promotion?

- What are the possible cross-effects of promotions on the product categories in the shopping basket?

This article is structured as follows: Section 2 introduces the data and the methodological framework used, which are divided into two main parts. The first part presents an initial insight into the effect of promotion on total household expenditure using a fixed-effect regression model. The second part, where the EASI demand system is used, studies how promotion affects budget allocation between the different categories of foods in the shopping basket. The main results are presented in Section 3, followed by a discussion and some concluding remarks in Section 4.

## 2. Materials and methods

### 2.1 Data

We performed the empirical analysis using consumer scanner panel data spinning from January 2017 to December 2017. The data were obtained from the Kantar Worldpanel Company's network in Spain and provided detailed shopping information from a representative random household sample from Spain. To identify and track products, each household was equipped with a barcode reader that allowed them to scan Universal Product Codes (UPCs) and report all the scanned information in their shopping ticket: purchase quantities, prices and a variable that referred to retailers' promotions, with the value of 1 when the household made a purchase under a promotion and 0 otherwise. The dataset also included some sociodemographic characteristics of the household: age, social class, and presence of children, region, and others.

To allow us to investigate the effects of promotions more closely, we selected only one retailer in only one region, knowing that promotions depend heavily on retailers' marketing strategies and that such strategies vary between retailers and regions. Our chosen retailer is in the region of Catalonia in northeast Spain. It has a 9.6% share of its market, making it the second largest company in its field in Spain [46]. Notably, this retailer sells more than 100,000 product lines in 173 stores. One of its main attributes is its implementation of promotional campaigns quite often yearly, more often than do other retailers [47].

The data are unbalanced panel data. We followed 280 households that stayed in the sample the entire year and made at least four shopping trips during this period, since more than half of the sample made at least four shopping trips. Our household selection process is presented in Fig 3.

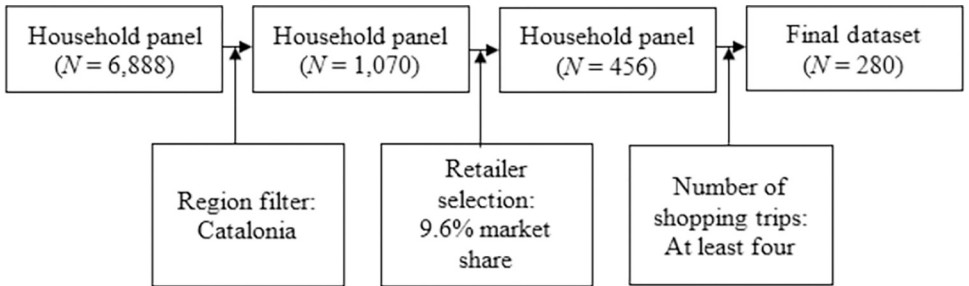

**Fig 3. Flowchart of the household selection process in this study.**

Household visits to the store were aggregated by week. As can be seen in Fig 4, around 12% of the households tended to do their shopping every 7 days, and more than 5% shopped every 6 days. In total, we had 3,564 observations of the sample.

Using the nutrition-based guidelines of the Ministry of Agriculture, Fisheries and Food in Spain, food products were aggregated into 13 categories: 1) grains and grain-based products; 2) vegetables and vegetable products; 3) starchy roots, tubers, legumes, nuts and oilseeds; 4) fruits, fruit products, and fruit and vegetable juices; 5) meat; 6) fish and other seafoods; 7) milk, milk products and milk imitation products; 8) cheese; 9) sugar, confectionery and pre-pared desserts; 10) composite dishes (animal and vegetable); 11) snacks and other foods; 12) drinks; and the 13) residual category. The categories' prices and the promotions' indices were computed following those in [42], using the weighted average of the prices paid by the house-holds for each product in each food category in each shopping trip. Detailed information on the categories and subproducts, together with the computation of the price and promotions indices, are presented in the (S1 Appendix).

It is important to mention that a proportion of the households did not buy certain products in a given week, resulting in zero values for the price variable. However, these zeros were

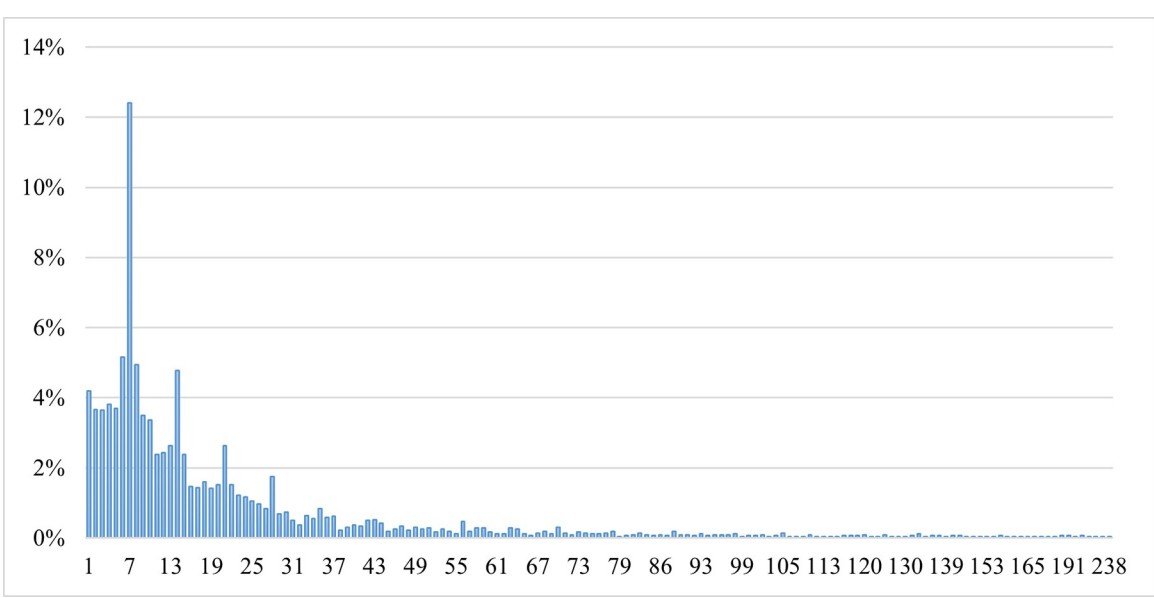

**Fig 4. Interpurchase time in days between shopping trips.**

**Table 1. Food category statistics.**

| Food category | Shopping frequency | Average quantity (kg/l) | Average price (€) | Mean budget share (%) | Percentage of trips with promotion (%) | Average expenditure without promotion in the shopping trip (€) | Average expenditure with promotion in the shopping trip (€) | Percentage change between expenditures (with & without promotion; %) |
|---|---|---|---|---|---|---|---|---|
| Grains & grain-based products | 15.10 (10.58) | 0.54 (0.81) | 0.89 (1.10) | 5.56 | 21.47 | 2.49 (2.27) | 4.85 (3.82) | 94.78 |
| Vegetables & vegetable products | 13.41 (11.23) | 0.90 (2.25) | 0.73 (1.31) | 3.89 | 29.56 | 3.02 (2.73) | 6.47 (5.32) | 114.24 |
| Starchy roots, tubers, legumes, nuts & oilseeds | 11.08 (9.47) | 0.24 (0.66) | 1.25 (2.63) | 1.97 | 24.73 | 2.85 (2.42) | 4.62 (4.28) | 62.11 |
| Fruits, fruit products, & fruit & vegetable juices | 16.35 (11.88) | 2.15 (3.61) | 0.42 (0.62) | 8.05 | 50.56 | 4.36 (4.47) | 7.63 (6.38) | 75.00 |
| Meat | 14.62 (10.60) | 0.52 (1.03) | 1.54 (2.10) | 11.81 | 32.63 | 10.83 (13.07) | 12.68 (16.19) | 17.08 |
| Fish & other seafoods | 12.61 (9.06) | 0.57 (1.11) | 2.26 (3.85) | 10.29 | 41.84 | 8.36 (7.94) | 17.44 (15.22) | 108.61 |
| Milk, milk products & milk imitation products | 14.99 (9.78) | 2.89 (5.29) | 0.54 (0.67) | 8.79 | 31.49 | 5.16 (5.27) | 8.51 (7.46) | 64.92 |
| Cheese | 13.25 (9.92) | 0.28 (0.53) | 1.67 (1.97) | 5.59 | 28.27 | 4.60 (4.37) | 8.38 (8.20) | 82.17 |
| Sugar, confectionery & prepared desserts | 14.51 (10.87) | 0.70 (1.09) | 1.40 (1.60) | 10.29 | 45.73 | 4.49 (3.98) | 8.60 (7.59) | 91.54 |
| Composite dishes (animal & vegetable) | 13.57 (10.71) | 0.64 (1.36) | 1.65 (2.92) | 7.24 | 43.41 | 5.57 (4.85) | 10.42 (7.85) | 87.07 |
| Snacks & other foods | 9.78 (8.66) | 0.12 (0.26) | 1.61 (2.18) | 2.48 | 46.85 | 2.18 (1.71) | 4.30 (2.63) | 97.25 |
| Drinks | 16.08 (11.04) | 6.63 (11.38) | 0.56 (1.23) | 14.21 | 49.95 | 6.87 (7.16) | 13.80 (13.48) | 100.87 |
| Residual category | 13.57 (10.71) | 0.77 (0.77) | 2.66 (4.52) | 9.82 | 36.69 | 7.00 (8.11) | 11.86 (12.50) | 69.43 |
| Expenditure per shopping trip | | | | | | 55.54 (40.16) | 58.82 (40.81) | 5.91 |

Source: Own elaboration based on Kantar Worldpanel data.

Note: Standard deviations are in parentheses.

replaced by the average values of the reported purchases of the other households in the sample (i.e., the households living in Catalonia).

Table 1 shows some descriptive statistics related to the 13 food categories. These statistics include the average number of shopping trips, the average shopping quantity, average price, mean budget share, expenditure per shopping trip, percentage of shopping trips with promotions, category expenditure when at least one product was on promotion and expenditures without any promotion.

As can be observed, on average, households tend to purchase 'Fruit, fruit products and fruit and vegetable juices', 'Drinks' and 'Grains and grain-based products' by a frequency of 16.35, 16.08 and 15.10 times, respectively. The least purchased categories are 'Snacks and other foods' and 'Starchy roots, tubers, legumes, nuts and oilseeds'.

Regarding quantities, 'Drinks' is the category that is purchased in the highest quantities (6.63 l), followed by 'Fruit, fruit products and fruit and vegetable juices' (2.15 kg). The smallest

quantities are observed for 'Snacks and other food' and 'Starchy roots, tubers Legumes, nuts and oilseeds'. We also see that 'Fish and seafood', with 'Meat', have the highest price (€2.26 and €1.54, respectively), while the lowest prices are observed for 'Fruit, fruit products and fruit and vegetable juices' and 'Milk, dairy products and milk product imitates'. The highest share of the household's budget is allocated to 'Drinks' (14.21%), followed by 'Meat' (11.81%), 'Fish and other seafood' (10.29%) and 'Sugar and confectionery and prepared desserts' (10.29%). A smaller budget is allocated to 'Starchy roots, tubers legumes, nuts and oilseeds' (1.97%) and 'Snacks and other food' (2.48%).

Concerning the percentage of shopping trips made by the households that included a promotion, it ranged from 21.47% for 'Grains and grain-based products' to 50.56% for 'Fruit, fruit products and fruit and vegetable juices'. For this same metric, we note that the percentage exceeds 40% for most food categories (drinks, snacks and other food, sugar and confectionery and prepared dessert, composite dishes, and fish and other seafoods). These high percentages of promotions lead to an increase in categories' expenditure, as shown in Table 1. Expenditures increase when promotions are used for all the categories. On average, households spend 55.54 € per shopping trip in the absence of promotion, and it increases by 5.91% when they buy at least once under promotion to get to 58.82€. However, such a rise is heterogeneous among food categories. For instance, 'Vegetables and vegetable products', 'Fish and other seafood' and 'Drinks' show the highest change in expenditure (114.24%, 108.61% and 100.87%, respectively), while the expenditure for 'Sugar and confectionery and prepared desserts', 'Grains and grain-based products' and 'Snacks' vary from 91.54% to 97.25%.

The sample sociodemographic characteristics are presented in Table 2. The households made 19.43 shopping trips annually on average. Most of the households live in Barcelona (78.93%), with the remainder distributed among the other provinces of Catalonia, similar to the distribution of the general population [48]. The average age of the household head is 51.59 years. About 58.22% of the households had 3–4 members, and 52.5% comprised couples with children. Of the children, 41.08% were aged 6–15 years. Most of the households were from the middle class, followed by the middle-high class.

## 2.2 Empirical model

In this section, we present the methodological framework of our analysis. Our methodological approach has two main parts, each of which addresses the research questions. The first part assesses the impact of prices and promotions on the total shopping food expenditure. The second part analyses the cross-effects of prices and promotions among the food categories by estimating a demand system and computing the price, promotion and expenditure elasticities.

**2.2.1 Effects of promotion on the shopping food expenditures.** To assess the impact of promotion and price on the total amount spent by the households on each shopping trip $(X_{ht})$, we used the following panel data regression model:

$$lnX_{ht} = \alpha_0 + \sum_{g=1}^{13} b_g lnp_{hgt} + \sum_{g=1}^{13} c_g Pm_{hgt} + r_t^{(h)}, \tag{1}$$

Where $r_t^{(h)} = \rho^{(h)} + u_t, u_{t\sim} i.i.d. \ N(0, \sigma_u^2)$.

$ln$ denotes the natural logarithm, since the model is presented under the log-log form, after which the price coefficient will be interpreted as 'elasticity'; $\alpha_0$, $b_g$ and $c_g$ are the regression coefficients related to each variable; $g$ is the food group number (1, 2,..., 13); $p_{hgt}$ and $Pm_{hgt}$ are the price and promotion indices, respectively. The symbol $u_t$ is the idiosyncratic error that varies over time, and $\rho^{(h)}$ is the time constant unobserved heterogeneity that represents the fixed-effect specification. According to [49], the fixed-effect model assumes the same slope

**Table 2. Profile of the households in the sample.**

| | Whole sample (N = 280) | | | | |
|---|---|---|---|---|---|
| **Variable** | **Mean or percent** | **Frequency (N)** | **Variable** | **Mean or percent** | **Frequency (N)** |
| Number of shopping trips | 19.43 | | Immigrant (%) | | |
| Household size (%) | | | Yes | 24.29 | 68 |
| 1 | 7.86 | 22 | No | 75.71 | 212 |
| 2 | 27.86 | 78 | Geographic area (%) | | |
| 3 | 29.29 | 82 | Barcelona | 78.93 | 221 |
| 4 | 28.93 | 81 | Girona | 9.64 | 27 |
| +5 | 6.07 | 17 | Lerida | 5.71 | 16 |
| | | | Tarragona | 5.71 | 16 |
| Life cycle (%) | | | Age of household head | 51.59 | |
| Single-parent households | 11.07 | 31 | Social class (%) | | |
| Independent youth | 1.79 | 5 | High | 15.36 | 43 |
| Adult couples without children | 14.64 | 41 | Middle | 46.07 | 129 |
| Young couples without children | 5.36 | 15 | Middle-high | 21.43 | 60 |
| Couples with middle-aged children | 23.57 | 66 | Middle-low | 17.14 | 48 |
| Couples with adult children | 14.64 | 41 | Children in household (%) | | |
| Couples with small children | 14.29 | 40 | Children 0–5 years | 14.29 | 40 |
| Independent adults | 3.21 | 9 | Children 6–15 years | 26.79 | 75 |
| Retired | 11.43 | 32 | No children | 58.93 | 165 |

Source: Own elaboration based on Kantar Worldpanel data.

and a constant variance across individuals; and since the household characteristics cannot vary over time, $\rho^{(h)}$ is allowed to be correlated with the other regressors. Then, the Hausman specification test was used to check which of the fixed-effect or random-effect models was the most appropriate for this analysis under the null hypothesis that individual effects are uncorrelated with any regressor in the model [49]. In our case, the null hypothesis of no correlation was rejected ($X^2_{0.000,26} = 157.92$), showing that the fixed-effect model is the most appropriate for the estimation (1).

**2.2.2 Cross-effect impact of promotions on budget allocation among food categories.** To analyse the effects of promotions on the demand for the 13 food categories, we specified the EASI demand model [45]. As already stated, the model allows for more flexible Engel curves and for interpreting the error terms as unobserved household preferences. The model is as follows:

$$w_{hgt} = \sum_{r=1}^{L} B_{gr} y_{ht}^{r} + \sum_{g=1}^{G} A_{ig} lnp_{hgt} + CPm_{hgt} + Dz_n + \varepsilon_{hgt} \tag{2}$$

for $h = 1, \cdots, H$; $g = 1, \cdots, G$; and $t = 1, \cdots, T$, where $w_{hgt}$ is the expenditure share of the food category $g$ for household $h$ in period $t$; $G$ is the number of food categories; $H$ is the number of households; $lnp_{hgt}$ is the vector of the logarithmic price indices; $y_{ht}$ is the log total real expenditure (the curvature of the Engel curve) which is calculated by deflating the nominal total

expenditure $X_{ht}$ with the Stone price index, as follows:

$$y_{ht} = lnX_{ht} - \sum_{g=1}^{G} lnp_{hgt}w_{hgt}, \tag{3}$$

$L$ is the highest degree of polynomials of $y_{ht}$ to be determined numerically; $Pm$ is a vector of the promotional indices; $z$ is an $n$ vector of the sociodemographic characteristics; $\varepsilon$ is the error term that captures the unobserved heterogeneity; and $A$, $B$, $C$ and $D$ are the coefficients to be estimated.

To determine the optimal degree of the polynomial ($L$) in (2), the model was estimated sequentially starting with $r = 1$ up to $r = 5$, and a joint test of the significance of the parameter $B_{gr}$ was performed. The $p$-value was $<0.00$ for $r = 3$, indicating that the cubic functional form was the most appropriate in this case to capture the curvature of the Engel curves.

As not all the households purchased food from all the categories in each shopping trip, zero purchases were relatively frequent in our data set. To account for the censored nature of the dependent variable, we followed the two-step procedure suggested by [50] even though criticised by [51] but it has been supported by [52] indicating that is one of most suitable methods to account for censoring. In the first step of this procedure, the decision to purchase is modelled as a dichotomous choice problem that has a value of 1 if a household consumes food in a category during a shopping trip and 0 otherwise. Then, the regression is used to compute the inverse Mills ratio (IMR) for each household, which is equal to the ratio of the probability density function to the cumulative distribution function of a distribution. The detailed model is presented in the supplementary material (S1 Appendix). Then, the IMR for each food category is used as an instrument in the second step of the demand estimation, as follows:

$$w_{hgt} = \sum_{g=1}^{G} A_{ig}lnp_{hgt} + \sum_{r=0}^{3} B_r y_{ht}^r + CPm_{hgt} + Dz_n + \delta R_{hgt} + \varepsilon_{hgt,} \tag{4}$$

Where, all the variables are the same as the ones stated in Eq (2), $R_{hgt}$ is the instrument referring to the calculated IMR.

Adding-up and homogeneity were imposed by the introduction of the following restrictions:

$$1_g^{'}b_0 = 1, 1_g^{'}b_r = 0 \ \forall r \neq 0, \tag{5}$$

And

$$1_g^{'}A = 1_g^{'}C = 0_g, 1_n^{'}D = 0_n, \tag{6}$$

Moreover, Slutsky symmetry is guaranteed by the symmetry of the $g \times g$ matrices of $A$.

**2.2.3 Endogeneity.** Two sources of endogeneity can arise from the demand system Eq (4). The first is related to the presence of budget shares in the Stone price index. This kind of endogeneity can be negligible, according to [53] and [45]. The second source of endogeneity is related to the real food expenditure ($y_{ht}$), which is a function of the nominal total expenditure ($X_{ht}$). To account for the latter, we followed [45] by using $\bar{y}_{ht}$ as an instrument for real food expenditure, which was estimated by replacing $w_{ht}$ with its mean, $\bar{w}_{ht}$, in the following equation:

$$\bar{y}_{ht} = lnX_{ht} - \sum_{g=1}^{G} lnp_{hgt}\bar{w}_g, \tag{7}$$

The model was estimated using the three-stage least squares (3SLS) method to account for endogeneity. The expenditure elasticities, Hicksian and Marshallian price elasticities with promotional elasticities, were estimated following [41,44,53], as described next.

The matrix for the compensated Hicksian price elasticities for a food category $g$ with respect to the price of food category $j$ is given by:

$$\epsilon = \bar{w}^{-1}(A) + \Omega\bar{w} - I, \tag{8}$$

where $\epsilon$ is $n \times n$ of the compensated demand elasticities, $\bar{w}$ is an identity matrix where the ones have been replaced by the mean of the budget shares, $A$ is the $n \times n$ matrix of price coefficients, $\Omega$ is an $n \times n$ matrix of ones, and $I$ is an identity matrix.

The vector of the expenditure elasticities ($\vartheta$) was calculated using the following equation:

$$\vartheta = \bar{w}^{-1}(I + \Theta'\mathrm{p})^{-1}\Theta + 1_g, \tag{9}$$

Where $\vartheta$ is a vector of $G \times 1$ of the estimated expenditure elasticities; $\Theta$ is a vector of expenditure semi-elasticity coefficients, which is $\sum_{r=0}^{3} rB_r y^{r-1}$; $p$ is a vector of the mean prices; and $1_j$ is a $G \times 1$ of ones.

The matrix of uncompensated Marshallian elasticities ($\theta$) was derived from the Slutsky equation using the following equation:

$$\theta = \epsilon - \bar{w}\vartheta, \tag{10}$$

The matrix of promotional elasticities, $\omega$, was derived as follows:

$$\omega = \bar{w}^{-1}(C) * \bar{P}_m, \tag{11}$$

Where $\bar{w}$ is an identity matrix where the ones have been replaced with the means of the budget shares; $C$ is the $n \times n$ matrix of promotional coefficients; and $\bar{P}_m$ is an identity matrix where the ones have been replaced by the mean of the promotional indices.

## 3. Results

### 3.1 Effects of prices and promotions on household shopping food expenditures

Table 3 shows the effects of prices and promotions on the households' food expenditures. The expenditure regression model was estimated with 3,564 observations, leading to an $R^2$ of 0.32. Considering that the regression model is expressed in log-log form, the price coefficients directly represent the elasticities. The values are less than unity, indicating the inelasticity of the categories. All the estimated price coefficients are positive and significant at the 1% level except for the 'Residual category'. The highest effects of price were observed for 'Snacks and other foods', 'Meat' and 'Fish and other seafoods' (0.125, 0.115 and 0.107, respectively). Leaving out the 'Residual category', the smallest elasticities were observed for 'Vegetables and vegetable products' (0.029), 'Starchy roots, tubers, legumes, nuts and oilseeds' (0.039) and 'Milk, milk products and milk imitation product' (0.043).

Similar to prices, promotions had a positive effect on the households' food expenditures. Except for the 'Residual category', the largest effects were found on 'Fish and seafoods' (0.858), 'Composite dishes' (0.806) and 'Fruit, fruit products, and fruit and vegetable juices' (0.802), consistent with the results shown in Table 1, considering that these categories had a significant share of the budget (7.24–10.29%) and had high promotion use (41.84–50.56%). The categories affected least by promotions are 'Grains and grain-based products' (0.359), 'Cheese' (0.400) and 'Starchy roots, tubers, legumes, nuts and oilseeds' (0.407).

**Table 3. Results of the estimation of the total expenditure regression.**

| Variable | Coeff | SE | Sig |
|---|---|---|---|
| Intercept | 3.286 | 0.037 | *** |
| *Prices* | | | |
| Grains & grain-based products | 0.051 | 0.013 | *** |
| Vegetables & vegetable products | 0.029 | 0.013 | ** |
| Starchy roots, tubers, legumes, nuts & oilseeds | 0.039 | 0.015 | *** |
| Fruit, fruit products, & fruit & vegetable juices | 0.092 | 0.015 | *** |
| Meat | 0.115 | 0.015 | *** |
| Fish and other seafoods | 0.107 | 0.017 | *** |
| Milk, milk products & milk imitation products | 0.043 | 0.015 | *** |
| Cheese | 0.104 | 0.021 | *** |
| Sugar, confectionery & prepared desserts | 0.072 | 0.011 | *** |
| Composite dishes (animal & vegetable) | 0.041 | 0.011 | *** |
| Snacks & other foods | 0.125 | 0.121 | *** |
| Drinks | 0.082 | 0.144 | *** |
| Residual category | 0.004 | 0.120 | |
| *Promotions* | | | |
| Grains & grain-based products | 0.359 | 0.124 | *** |
| Vegetables & vegetable products | 0.430 | 0.144 | *** |
| Starchy roots, tubers, legumes, nuts & oilseeds | 0.407 | 0.117 | *** |
| Fruit, fruit products, & fruit & vegetable juices | 0.802 | 0.141 | *** |
| Meat | 0.694 | 0.140 | *** |
| Fish & other seafoods | 0.858 | 0.112 | *** |
| Milk, milk products & milk imitation products | 0.602 | 0.107 | *** |
| Cheese | 0.400 | 0.129 | *** |
| Sugar, confectionery & prepared desserts | 0.705 | 0.126 | *** |
| Composite dishes (animal & vegetable) | 0.806 | 0.113 | *** |
| Snacks & other foods | 0.125 | 0.098 | |
| Drinks | 0.655 | 0.081 | *** |
| Residual category | 1.046 | 0.099 | *** |
| *N* | 3564 | | |
| $R^2$ | 0.326 | | |

Source: Own elaboration based on Kantar Worldpanel data.

Notes: Coeff: Coefficient relative to the price and promotion variables, SE: standard error, Sig: Significance level.

***, **, * indicate significance at $p < 0.001$, $p < 0.05$ and $p < 0.01$, respectively.

## 3.2 Price, expenditure and promotional elasticities

After we estimated the effects of changes in prices and promotions on food expenditures, we focused on the effect of total food expenditure, prices and promotions on the demand for each food category that was considered in this study. Table 4 shows the average uncompensated own-price, cross-price and expenditure elasticities. All own-price elasticities were negative and significant at the 5% level, except for that of 'Starchy roots', which was negative but insignificant. Moreover, except for the 'Residual category', which includes a wide range of quite heterogeneous products, all food categories were price-inelastic. Among these inelastic categories, 'Drinks' showed the highest own-price elasticity (-0.851), followed by 'Milk, milk products and milk imitation products' (-0.701) and 'Sugar, confectionery and prepared desserts' (-0.667).

**Table 4. Mean uncompensated elasticities and expenditure elasticities.**

| Food category | Grains | Vegetables | Starchy | Fruits | Meat | Fish | Milk | Cheese | Sugar | Composite | Snacks | Drinks | Residual |
|---|---|---|---|---|---|---|---|---|---|---|---|---|---|
| Grains | **-0.627** | -0.044 | -0.025 | -0.001 | -0.111 | -0.007 | -0.053 | -0.116 | -0.099 | -0.101 | -0.061 | -0.086 | 0.096 |
| | (0.061) | (0.018) | (0.013) | (0.026) | (0.035) | (0.034) | (0.029) | (0.027) | (0.035) | (0.027) | (0.017) | (0.035) | (0.062) |
| Vegetables | -0.041 | **-0.522** | -0.038 | -0.059 | -0.018 | -0.055 | -0.034 | 0.003 | -0.018 | 0.058 | -0.013 | 0.044 | -0.148 |
| | (0.023) | (0.058) | (0.015) | (0.030) | (0.035) | (0.036) | (0.034) | (0.030) | (0.029) | (0.027) | (0.019) | (0.033) | (0.077) |
| Starchy | -0.011 | -0.049 | **-0.226** | -0.027 | -0.085 | -0.055 | -0.046 | -0.095 | -0.130 | -0.089 | -0.045 | -0.007 | 0.692 |
| | (0.036) | (0.028) | (0.147) | (0.036) | (0.046) | (0.042) | (0.047) | (0.043) | (0.048) | (0.050) | (0.041) | (0.040) | (0.157) |
| Fruits | 0.014 | -0.034 | -0.022 | **-0.608** | 0.017 | -0.115 | 0.005 | -0.030 | -0.018 | -0.034 | -0.032 | -0.029 | -0.081 |
| | (0.016) | (0.015) | (0.008) | (0.037) | (0.024) | (0.022) | (0.019) | (0.015) | (0.021) | (0.018) | (0.010) | (0.018) | (0.047) |
| Meat | -0.051 | -0.020 | -0.034 | -0.007 | **-0.628** | -0.053 | -0.003 | -0.036 | -0.047 | -0.033 | -0.037 | -0.039 | -0.217 |
| | (0.015) | (0.010) | (0.007) | (0.016) | (0.031) | (0.020) | (0.017) | (0.015) | (0.017) | (0.014) | (0.008) | (0.020) | (0.042) |
| Fish | 0.009 | -0.027 | -0.027 | -0.094 | -0.039 | **-0.484** | -0.030 | -0.060 | -0.036 | -0.053 | -0.071 | 0.010 | -0.111 |
| | (0.017) | (0.013) | (0.008) | (0.017) | (0.023) | (0.039) | (0.019) | (0.016) | (0.019) | (0.017) | (0.011) | (0.019) | (0.047) |
| Milk | -0.026 | -0.025 | -0.029 | -0.006 | 0.009 | -0.044 | **-0.701** | -0.034 | -0.023 | 0.003 | -0.050 | -0.042 | -0.133 |
| | (0.016) | (0.013) | (0.010) | (0.018) | (0.025) | (0.022) | (0.039) | (0.017) | (0.021) | (0.019) | (0.010) | (0.023) | (0.048) |
| Cheese | -0.090 | 0.005 | -0.045 | -0.028 | -0.025 | -0.087 | -0.025 | **-0.218** | -0.012 | 0.002 | -0.042 | -0.013 | -0.199 |
| | (0.024) | (0.020) | (0.015) | (0.022) | (0.032) | (0.029) | (0.026) | (0.053) | (0.028) | (0.024) | (0.016) | (0.024) | (0.067) |
| Sugar | -0.026 | -0.003 | -0.036 | 0.004 | 0.001 | -0.008 | 0.012 | -0.004 | **-0.667** | -0.026 | -0.018 | -0.006 | 0.037 |
| | (0.015) | (0.010) | (0.009) | (0.016) | (0.021) | (0.019) | (0.018) | (0.015) | (0.036) | (0.017) | (0.008) | (0.018) | (0.039) |
| Composite | -0.049 | 0.036 | -0.035 | -0.017 | 0.004 | -0.045 | 0.038 | 0.005 | -0.034 | **-0.495** | -0.027 | 0.001 | -0.097 |
| | (0.020) | (0.014) | (0.014) | (0.020) | (0.024) | (0.024) | (0.022) | (0.019) | (0.024) | (0.036) | (0.012) | (0.021) | (0.052) |
| Snacks | -0.090 | -0.003 | -0.040 | -0.057 | -0.081 | -0.231 | -0.115 | -0.073 | -0.039 | -0.056 | **-0.435** | -0.020 | 0.839 |
| | (0.037) | (0.029) | (0.032) | (0.030) | (0.045) | (0.045) | (0.045) | (0.039) | (0.040) | (0.035) | (0.134) | (0.041) | (0.140) |
| Drinks | -0.020 | 0.006 | -0.017 | -0.019 | -0.008 | 0.009 | -0.017 | -0.017 | -0.031 | -0.020 | -0.018 | **-0.851** | 0.006 |
| | (0.009) | (0.007) | (0.005) | (0.011) | (0.016) | (0.014) | (0.012) | (0.010) | (0.013) | (0.010) | (0.005) | (0.028) | (0.027) |
| Residual category | 0.037 | -0.086 | 0.112 | -0.112 | -0.300 | -0.170 | -0.158 | -0.156 | -0.044 | -0.132 | 0.183 | -0.069 | **-1.461** |
| | (0.034) | (0.029) | (0.033) | (0.036) | (0.052) | (0.047) | (0.043) | (0.039) | (0.039) | (0.036) | (0.039) | (0.041) | (0.204) |
| *Expenditures* | **1.235** | **0.840** | **0.173** | **0.968** | **1.204** | **1.013** | **1.099** | **0.779** | **0.740** | **0.714** | **0.402** | **0.998** | **1.538** |
| | (0.224) | (0.155) | (0.223) | (0.059) | (0.071) | (0.055) | (0.103) | (0.081) | (0.077) | (0.065) | (0.257) | (0.065) | (0.138) |

Source: Own elaboration based on Kantar Worldpanel data.

Notes: Bootstrapped standard errors (500 replications) are in parentheses. The numbers in bold in the diagonal and horizontal lines refer to, own-price elasticities and expenditure elasticities respectively. Regarding the promotion elasticities, the results are presented in Table 5. Except for the 'Residual category', all the food categories had positive own-promotion elasticities that were significant at the 5% level. Even though the values of the own-promotion coefficients were relatively small, those of 'Composite dishes', 'Sugar, confectionery and prepared desserts', 'Fish and other seafoods' and 'Drinks' showed the highest values (0.177, 0.171, 0.158 and 0.132, respectively). These elasticities are consistent with the descriptive statistics shown in Table 1 (which indicates that these categories have a significant share of promotion efforts) and with the results shown in Table 3 (where we observed that promotions for the listed categories had strong effects on the total expenditure). In contrast, promotions for 'Starchy roots, tubers legumes, nuts and oilseeds' and 'Meat' had the lowest impacts (0.052 and 0.062, respectively).

'Cheese' and 'Snacks and other foods' had lower price elasticities (-0.218 and -0.435, respectively).

As for cross-price elasticities, many of these were quite low and insignificant. Moreover, the most significant values revealed complementary relationships (i.e., negative cross-price elasticities). The only exceptions were the relationships between 'Vegetables and vegetable products' with respect to 'Composite dishes' and 'Snacks and other foods' with respect to the 'Residual category', which were positive and significant, indicating a substitution relationship. In any case, the cross-price elasticities were much smaller in absolute value relative to the own-price elasticities.

The expenditure elasticities were positive and significant at the 5% level, except for that of the 'Starchy roots, tubers, legumes, nuts and oilseeds', which was positive but insignificant. The results indicate that all the food categories behaved as normal goods. Except for the 'Residual category', the most expenditure-elastic food categories were 'Grains and grain-based

**Table 5. Promotion elasticities.**

| Food category | Grains | Vegetables | Starchy | Fruits | Meat | Fish | Milk | Cheese | Sugar | Composite | Snack | Drinks | Residual |
|---|---|---|---|---|---|---|---|---|---|---|---|---|---|
| Grains | **0.114** (0.017) | -0.006 (0.004) | -0.002 (0.002) | -0.008 (0.008) | -0.021 (0.009) | -0.050 (0.015) | 0.030 (0.015) | -0.004 (0.005) | -0.008 (0.013) | -0.019 (0.010) | 0.009 (0.006) | -0.057 (0.023) | 0.031 (0.040) |
| Vegetables | -0.028 (0.017) | **0.123** (0.018) | 0.003 (0.005) | 0.003 (0.016) | -0.028 (0.015) | -0.004 (0.019) | -0.013 (0.021) | -0.003 (0.010) | -0.060 (0.019) | -0.006 (0.015) | -0.007 (0.014) | -0.101 (0.043) | -0.061 (0.041) |
| Starchy roots | -0.009 (0.018) | -0.012 (0.011) | **0.052** (0.023) | -0.031 (0.026) | -0.054 (0.026) | -0.052 (0.041) | -0.017 (0.029) | -0.013 (0.015) | -0.027 (0.027) | 0.007 (0.030) | -0.034 (0.018) | -0.028 (0.061) | 0.069 (0.055) |
| Fruits | -0.008 (0.005) | 0.008 (0.003) | -0.002 (0.002) | **0.129** (0.016) | -0.014 (0.011) | -0.014 (0.010) | -0.010 (0.013) | 0.001 (0.004) | -0.010 (0.009) | -0.009 (0.011) | -0.001 (0.004) | -0.094 (0.023) | 0.026 (0.026) |
| Meat | -0.002 (0.004) | -0.005 (0.002) | -0.004 (0.001) | -0.022 (0.004) | **0.062** (0.011) | -0.015 (0.007) | -0.010 (0.006) | 0.005 (0.004) | -0.005 (0.008) | 0.004 (0.008) | -0.006 (0.003) | -0.046 (0.011) | 0.078 (0.027) |
| Fish | -0.014 (0.003) | -0.003 (0.002) | -0.003 (0.001) | -0.011 (0.004) | -0.013 (0.004) | **0.158** (0.014) | -0.020 (0.005) | -0.008 (0.002) | -0.026 (0.006) | -0.017 (0.004) | -0.004 (0.003) | -0.067 (0.010) | -0.035 (0.029) |
| Milk | -0.005 (0.003) | 0.002 (0.003) | 0.002 (0.002) | -0.010 (0.005) | -0.020 (0.005) | -0.022 (0.005) | **0.126** (0.011) | 0.002 (0.003) | -0.022 (0.006) | -0.015 (0.005) | -0.001 (0.003) | -0.014 (0.017) | -0.019 (0.027) |
| Cheese | -0.015 (0.011) | -0.009 (0.004) | -0.004 (0.003) | -0.012 (0.009) | -0.011 (0.013) | 0.011 (0.016) | -0.011 (0.012) | **0.092** (0.017) | -0.026 (0.012) | -0.030 (0.011) | 0.001 (0.007) | -0.048 (0.025) | -0.059 (0.035) |
| Sugar | -0.013 (0.006) | -0.009 (0.002) | -0.004 (0.001) | -0.018 (0.004) | -0.012 (0.006) | -0.019 (0.008) | -0.013 (0.009) | -0.008 (0.003) | **0.171** (0.018) | -0.013 (0.008) | -0.007 (0.004) | -0.050 (0.013) | -0.076 (0.025) |
| Composite | -0.010 (0.006) | -0.008 (0.003) | -0.002 (0.002) | -0.021 (0.006) | -0.023 (0.008) | -0.009 (0.009) | -0.024 (0.008) | -0.010 (0.004) | 0.007 (0.009) | **0.177** (0.016) | -0.007 (0.004) | -0.032 (0.016) | -0.063 (0.027) |
| Snacks | -0.005 (0.009) | -0.015 (0.005) | -0.004 (0.004) | -0.041 (0.012) | -0.025 (0.015) | -0.004 (0.018) | -0.056 (0.017) | -0.012 (0.007) | 0.024 (0.002) | -0.004 (0.016) | **0.084** (0.024) | 0.041 (0.044) | 0.031 (0.044) |
| Drinks | -0.004 (0.002) | -0.004 (0.001) | -0.001 (0.001) | -0.011 (0.002) | -0.014 (0.003) | -0.014 (0.004) | -0.009 (0.003) | -0.005 (0.002) | -0.002 (0.003) | -0.005 (0.003) | 0.000 (0.001) | **0.132** (0.017) | 0.091 (0.023) |
| Residual | 0.012 (0.015) | -0.013 (0.008) | 0.007 (0.006) | 0.017 (0.018) | 0.063 (0.022) | -0.033 (0.027) | -0.016 (0.023) | -0.021 (0.012) | -0.078 (0.025) | -0.045 (0.019) | 0.009 (0.013) | 0.195 (0.050) | **0.049** (0.047) |

Source: Own elaboration based on Kantar Worldpanel data.

Notes: Bootstrapped standard errors (500 replications) are in parentheses. The numbers in bold in the diagonal line refer to, own-promotion elasticities.

products' (1.235), 'Meat' (1.204), 'Milk, milk products and milk imitation products' (1.099) and 'Fish and other seafoods' (1.013). The expenditure-inelastic categories were 'Snacks and other foods' (0.402) and 'Composite dishes' (0.714).

For the cross-promotion effects, most were negative, except for the positive effect of 'Vegetables and vegetable products' on 'Fruit, fruit products, and fruit and vegetable juices' (0.008). As can be observed from Table 5, promotions for 'Vegetables and vegetable products' negatively affected 'Snacks and other foods' (-0.015), 'Sugar, confectionery and prepared desserts' (-0.009) and 'Drinks' (-0.004). Promotions for 'Fruit, fruit products, and fruit and vegetable juices' decrease the demand for 'Milk, milk products and milk imitation products' (-0.010), 'Sugar, confectionery and prepared desserts' (-0.018), 'Snacks and other foods' (-0.041) and 'Drinks' (-0.011). Promotions for 'Sugar, confectionery products and prepared desserts' decreased the demand for 'Vegetables and vegetable products' (-0.060), 'Milk, milk products and milk imitation products' (-0.022) and 'Cheese' (-0.026). Promotions for 'Composite dishes' reduced the demand for 'Cheese' (-0.030) and 'Fish and other seafoods' (-0.017), but promotions for 'Fish and other seafoods' negatively affected 'Meat' (-0.015). Promotions for the 'Meat' category negatively affected the 'Composite dishes' (-0.023) and 'Fish and other seafoods' (-0.013). Promotions for 'Drinks' decreased the demand for most of the food categories, except for 'Starchy roots, tubers legumes, nuts and oilseeds', 'Milk, milk products and milk imitation products', 'Composite dishes' and 'Snacks and other foods'.

In general, the cross-promotion effects were smaller than the own-promotion effects, as in the case of the prices. Moreover, we observed that the cross-promotion effects were highly asymmetric between the food categories. For example, the effect of 'Vegetables and vegetable products' on 'Sugar, confectionery and prepared desserts' was -0.009, whereas the effect of 'Sugar, confectionery and prepared desserts' on 'Vegetables and vegetable products' was much higher (-0.060). Another example is the effect of 'Meat' on 'Fish and other seafoods', which was lower (-0.013) than the other way around (-0.015).

## 4. Discussion

In this study, we present estimates of the impacts of both price and promotion on household expenditures for all items in the shopping basket, as well as on the allocation of the shopping budget across the shopping basket items. To do so, we used two complementary quantitative tools. First, we estimated a fixed-effect regression model to study the impact of price and promotion on household expenditures in our selected retailer outlets. Second, we used the censored EASI demand system established by [45] to calculate the expenditure, price and promotion elasticities in order to analyse the cross-product effects of price and promotion, considering the censored nature of the dependent variable, which, in this study, was the expenditure share.

The results of the fixed-effect regression indicate that both price and promotion had a positive effect on the household expenditures. The price elasticities were lower than unity, indicating the price-inelastic nature of the food products and that the households were not price-sensitive. This result further indicates that the food products were weakly separable from each other. This result is in line with that of [43] and in contrast with the result of [42], which showed a negative effect of price on household expenditures and more price-sensitive consumers. These contrasting results are probably due to the different countries studied and the difference in the product aggregation, which, in [42], was focused on the retailers' product assortment, and in the current study, was based on the Spanish nutrition guidelines.

As for promotion, various studies [41,54] reported its positive impact on household expenditures, as explained by the retailers' promotional objective of enhancing their establishments' sales in other words, pushing households to spend more while shopping.

The results of the EASI demand system showed that demand for all the food categories, except for the 'Residual category', was inelastic in response to price changes, similar to [55,56] in Vietnam, [57] in the USA and [58] in Spain. 'Drinks' showed the highest own-price elasticity, as mentioned in [43,59]. Contrary to the findings of [58,60] in Spain, 'Milk, milk products and milk imitation products' had the second highest elasticity, similar to [42,59,61,62]. In any case, these comparisons are not straightforward, as the food aggregations differ across studies.

As for the cross-price elasticities in this study, many of them were quite low and insignificant. Moreover, the most significant cross-price elasticities revealed complementary relationships, as we expected, considering the broad number of food categories we considered. Substitution relationships were expected in each food category, while complementarity or even independent relationships were expected between categories, in line with [57]. In any case, the cross-price elasticities were much smaller in absolute value relative to the own-price elasticities, as in [63].

From the expenditure elasticities, we saw that all the food categories behaved as normal goods, which is also quite consistent with our expectations, as we are dealing with food categories and not with specific food products, among which we could expect some inferior goods behaviour. After the 'Residual category', the 'Grains and grain-based products', 'Meat', 'Milk, milk products and milk imitation products' and 'Fish and other seafoods' categories had the

highest sensitivity to expenditure changes, as they were elastic. These results are supported by other studies in Spain [59,60] for 'Milk, milk products and milk imitation products' and by [58,62,64] for 'Meat' and 'Fish and other seafoods'. Other studies outside Spain reported similar findings [56,65,66].

Promotion had a positive effect on the household expenditures in this study, even though the values were relatively small. The values for 'Composite dishes', 'Sugar, confectionery and prepared desserts', 'Fish and other seafoods' and 'Drinks' were the highest. These results are quite similar to those of [43], that in which promotion had the strongest effect on unhealthy products, such as beverages, soft drinks and confectionery, while the elasticity of the 'Meat' category was among the smallest across the categories. In any case, the values we obtained were higher. Differences in the econometric model, the country of study or the aggregation level of food products could have caused such discrepancies.

The negative spillover effects of promotions is the predominant pattern in this study, as many of the cross-promotion coefficients were negative, which indicates that an increase in promotion in one category decreased the demand for the other categories of the shopping basket. This behaviour is quite intuitive; as promotion induces a higher demand for the promoted product, households will switch to the non-promoted category with the promoted one, which will provoke a small but significant budget allocation between categories in the shopping basket. However, we also found a statistically significant indication of a positive spillover effect of the promotion of 'Fruit, fruit products, and fruit and vegetables juices' on the demand for 'Vegetables and vegetables products'. These cross-effects are asymmetric between related categories, as [35] also found, although in a non-food context where they studied the effects of price changes and promotions on grocery categories (i.e., laundry detergents, fabric softeners, cake mixes and cake frostings). Similar findings have been reported by [67] with [68] for soft drinks and coffee respectively. These studies reported that the own-promotion effects were stronger than cross-effects and that there was an asymmetric promotion cross-effect between pairs of categories.

Our results can contribute to improving retailers' understanding of consumers' reactions to promotion strategies for different food categories in Spain. While retailers analyse the impact of their promotions' strategies on a specific promoted food product, cross-effects on non-promoted categories are not considered, which can be negative. Although such negative effects are of lower magnitude than the own effect, the accumulated negative effect on the rest of the categories should not be underestimated. That is, retailer managers in Catalonia should promote categories such as 'Fish and other seafoods', 'Composite dishes', 'Fruits, fruit products, and fruit and vegetable juices', 'Sugar, confectionery and prepared desserts' and 'Drinks' with caution as to the cross-effects.

From a health perspective, although this was not the focus of this study, our results indicate that promotions for healthier food categories, such as 'Fruit, fruit products, and fruit and vegetable juices' and 'Vegetables and vegetable products' had a negative effect on 'Snacks and other foods', 'Sugar, confectionery and prepared desserts' and 'Drinks'. However, the reverse effect was also seen; that is, promotions for 'Drinks' decreased the budget share of most of the categories, including of 'Fruit, fruit products, and fruit and vegetable juices' and 'Vegetables and vegetable products'.

In any case, our results should be interpreted with caution, as they refer to a specific geographical area and to a specific retailer chain with heterogeneous food categories.

In our research paper, it is crucial to acknowledge certain limitations that may affect the scope and applicability of our findings. One significant limitation of our paper is the reliance on data collected from 2017 that may not accurately reflect the current state of markets, trends, and behaviors that could have been changed especially after COVID 19. Additionally,

extending this study to other areas and retailers with different business profiles could be challenging. Furthermore, our dataset reported promotions as a categorical variable, therefore we couldn't estimate their intensities.

While our study has provided valuable contributions to the current understanding of the topic at hand, there remain several promising directions for future research. From a food policy perspective, it could be interesting to differentiate between promotions for healthy and unhealthy products and to check if promotions for food products in each category contribute to a healthier or unhealthier diet, considering the total food expenditure. Either way, this potential future research should start with a proper definition of what is meant by 'healthy' and 'unhealthy' and with a reclassification of food categories according to this distinction. And by using recent data, we can delve into the consumer reactions towards sales promotions during COVID-19 and afterwards.

## Supporting information

**S1 Appendix. Supplementary appendix.**
(DOCX)

## Author Contributions

**Conceptualization:** Wafa Mehaba, Djamel Rahmani, José Maria Gil.

**Data curation:** Wafa Mehaba.

**Formal analysis:** Wafa Mehaba, Djamel Rahmani.

**Investigation:** Djamel Rahmani.

**Methodology:** Wafa Mehaba, Djamel Rahmani, José Maria Gil.

**Supervision:** Djamel Rahmani, José Maria Gil.

**Validation:** José Maria Gil.

**Writing – original draft:** Wafa Mehaba.

**Writing – review & editing:** Wafa Mehaba, Djamel Rahmani, José Maria Gil.

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
