## [Decision Letter · Decision Letter 0]

23 Jan 2024

PONE-D-23-33462The allocation of household food budget among shopping basket items: How is it influenced by promotions?PLOS ONE

Dear Dr. Mehaba,

Thank you for submitting your manuscript to PLOS ONE. After careful consideration, we feel that it has merit but does not fully meet PLOS ONE’s publication criteria as it currently stands. Therefore, we invite you to submit a revised version of the manuscript that addresses the points raised during the review process.All reviewers have agreed that your manuscript needs substantial improvement before further consideration. After addressing reviewers' comments, it is highly recommended to have your paper proofread by professional English proofreader.

We look forward to receiving your revised manuscript.

Kind regards,

Mohammed Al-Mahish, Ph.D.

Academic Editor

PLOS ONE

Journal Requirements:

Reviewers' comments:

Reviewer's Responses to Questions

**Comments to the Author**

1. Is the manuscript technically sound, and do the data support the conclusions?

Reviewer #1: Partly

Reviewer #2: Partly

Reviewer #3: Yes

2. Has the statistical analysis been performed appropriately and rigorously? 

Reviewer #1: Yes

Reviewer #2: Yes

Reviewer #3: Yes

3. Have the authors made all data underlying the findings in their manuscript fully available?

Reviewer #1: No

Reviewer #2: Yes

Reviewer #3: No

4. Is the manuscript presented in an intelligible fashion and written in standard English?

Reviewer #1: No

Reviewer #2: Yes

Reviewer #3: Yes

5. Review Comments to the Author

Reviewer #1: Dear Editors

Thank you for giving me the opportunity to review the manuscript, titled “The allocation of household food budget among shopping basket items: How is it influenced by promotions?”

The authors of the study chose a very important topic related to household food expenditure and the findings may have the potential to influence public health nutrition policies and interventions.

The study aimed to examine the effects of retail promotions on household food expenditure and to assess whether promotions affect reallocation of food shopping budget. The authors used data from a chain of supermarkets in Catalonia (Spain), using micro-panel scan data from Kantar World Panel for 2017, and conducted data analysis to address the three research questions: (i) to what extent do promotion and price influence households to change their food shopping expenditures? (ii) which food category has the strongest effect on food shopping expenditures when sold under promotion? and (iii) what are the cross-effects of promotion on food categories of the household food shopping basket?

The authors conducted the data analysis in two steps: An expenditure regression model to predict the effect of promotion on household food expenditures and a censored Exact Affine Stone Index (EASI) to estimate the promotion own and cross-effect, and found that promotion had a positive own-effect and mostly a negative asymmetric cross-effect. implying a small but significant budget reallocation.

I found the topic of the study interesting and the findings with potential to impact public health nutrition. However, the authors need to address the major and minor issues before the paper can be considered for publication.

Major issues

The paper has not been written in an organised way; therefore, the authors need to structure the paper under these sections (or follow the PLOS ONE guidelines): abstract, introduction, methods (study population, theoretical framework, and statistical method), ethics, results, discussion, conclusion, and references.

The introduction section needs to end up with identifying the research gap, study objective(s), followed by the three research questions.

The study population can be presented by a flowchart, showing number of households and/or household purchases, with exclusion criteria. Currently, it is not clear whether the authors included alcoholic beverages as part of household food purchases (I don’t think alcoholic beverages can be included as part of household food basket or purchases). What other food/beverage products need to be excluded (e.g., products purchased during Eister or Christmas), or weekly household purchases less than a certain amount of money or some other products that the authors excluded or may exclude them (e.g., foods for pets).

A theoretical framework (or conceptual framework) can be useful to guide the conceptual relationships between promotions and price and the outcome of interest, in line with the study objective(s) and/or research questions. Also, the authors need to provide a definition for retail promotions (I guess promotion was self-reported by households in this study).

Statistical methods: The outcome of interest, predictors, and statistical models (the expenditure regression, and EASI) need to be fully described. Some formulas, which are not in line with the research questions, can be moved to supplementary materials (e.g., the formula on weighted average price). In the regression model, variables on socioeconomic status and demographic of households need to be included, in addition to the variables on promotion and price, because currently the expenditure regression does not account for such important explanatory variables (for example, household food expenditures can be affected by household size).

The results should present baseline characteristics of households as well as frequency of household food purchases (by food categories, prices, quantities, etc.), followed by results in response to each of the research questions in tables/figures that are well organised and clearly presented (the authors can have a look at the published papers for examples of tables and figures). The authors should use terms that are easily understood and are meaningful to a general audience of the paper (all readers are not econometricians to understand coefficients, significance shown by ***, or other technical jargons).

Discussion: The first paragraph may cover an overall summary of the findings in the study, followed by comparison of the findings with relevant literature and possible explanations for discrepancies and similarities in the findings, with references. The study limitations and strengths, and the findings’ significance on public health nutrition policies can be discussed. Further research highlighted.

Conclusion: The overall interpretation of the findings in the context of Catalonia (or Spain) or wider region and world can be described. Recommendations may be suggested considering the findings.

Minor issues

English language needs substantial improvement throughout the paper.

There are technical jargons that the authors may need to provide a brief definition of them whenever they are used the first time in the paper (e.g., micro-panel in line 16).

The authors used some unnecessary words in some parts of the paper which can be confusing. For example, from lines 132-133 “Thus, three major research questions which are worth to be answered represent the objectives of our study.”. The authors can use a clear and concise sentence, instead. For example, they can use “In line with the study objectives, three research questions were examined.”

What is shopping ticket expenditure? (for example, in line 279 and elsewhere). Please use key terms consistently throughout the paper, as currently it is not clear whether “shopping ticket expenditure” means “shopping food expenditure” or something else.

The authors stated that zero purchases are relatively frequent in their data sets (lines 317-318). Is there a way that the authors can use to avoid “zero purchases”, because currently it seems (from the way the authors coded household purchases) that it was assumed that a household would purchase all food categories in each shopping trip (lines 317-318) which does not seem logical.

There is a contradiction in the findings in lines 405-407 “…except for the ‘Residual category’, which includes a wide range of quite heterogeneous products, all food categories are price inelastic.” with those in line 408 “As can be observed, the category of ‘Drinks’ shows the highest own-price elasticity…”

The authors need to check such issues throughout the paper.

Reviewer #2: This interesting study authors investigated the impact of retail promotions on total shopping expenditure and whether they lead to a shift in the shopping budget. The results showed a positive own effect and a predominantly negative asymmetric cross effect, which indicates a remarkable but small budget shift.

The outline of the paper is generally appropriate and has some strengths, although some major details need to be revised before acceptance:

1. The overarching objective of the study is to evaluate the impact of retail promotions on the overall expenditure in shopping baskets. However, the introduction section currently narrows its focus to a specific aspect, overlooking other potential effects that advertisements may exert on consumers. This omission hinders a comprehensive review of relevant studies and their implications. It's important to note that the effectiveness of retail promotions can vary based on factors such as the type of promotion, the target audience, market conditions, and the overall economic environment.

2. The research methodology effectively elucidates various intricacies; nevertheless, a crucial point of contention lies in the justification of utilizing dated data from 2017. The authors must substantiate how the household food budget, consumer behavior, and advertising methods have remained unchanged since then. This is essential to ensure the article's relevance and appeal to contemporary readers. It also needs to be mentioned as a limitation of the research at the end of the conclusion.

3. This concern extends to the references, as a substantial portion of them pertains to older works. To enhance the article's current appeal and scholarly robustness, it is essential to incorporate recent research findings and participation, thereby fortifying the foundation of the study with up-to-date and pertinent insights.

Reviewer #3: This is a good paper. I think the analysis is sound and general approach is sound and appropriate. The authors addressed the objectives of the paper with commonly used methods, addressing concerns of endogeneity and other issues to my satisfaction. The paper is well written and has clearly stated objectives. Well done.

I do have some minor comments and questions:

I do not think it is clear that food retail competition has "increased" as the authors claim in several places in the paper. If anything food retail has consolidated, reducing concentration in some countries. I would like the authors to support this claim with an example or statistic like a CR4 ratio or HHI data. I suspect that competition has declined and that in an environment with fewer firms, differentiation is the optimal strategy.

I am assuming, these "promotions" are not price reductions but rather advertisement and similar marketing activities. Can you measure the "intensity" of these marketing events somehow? currently it looks like you are using a yes/no variable. I would like to see the intensity reflected, if possible.

Line 234 and other places have an error message. Also, check equation numbers, they may be off (did you skip 7?) Double check your math notation and order of terms, especially eq. 6 and 8.

The specification in equation 5 makes sense to me. One question I have is about the potentially confounding effect of time. You have done well to include individual fixed-effects. Is there a reason not to include some sort of time based fixed-effect? Perhaps there is a seasonal component to expenditures that you are not accounting for--such a holiday season where both promotions and holiday traditions increase expenditure. You would want to disentangle these effects.

Line 325 and 343: I interpreted "instrument" to mean "proxy", an important distinction in my vocabulary. Did you use a true IV approach?

This is perhaps a thought for the future, but it occurred to me while reading your manuscript. Is there a way to connect your first and second models? To me, you measure first how promotions impact expenditure and then how promotions impact the expenditure share. I believe they may be related. Could equation 5 be a first stage for the EASI model? Perhaps you could isolate the effects of promotional impacts on expenditure share via the impact on expenditure. These are rough thoughts, but I hope they are helpful.

6. PLOS authors have the option to publish the peer review history of their article (what does this mean?). If published, this will include your full peer review and any attached files.

Reviewer #1: **Yes: **Essa Tawfiq

Reviewer #2: **Yes: **Sina Ahmadi Kaliji

Reviewer #3: **Yes: **Andrew E Anderson

---

## [Author Response · Author response to Decision Letter 0]

15 Mar 2024

Major concerns

Reviewer#1

Concern # 1: The paper has not been written in an organized way; therefore, the authors need to structure the paper under these sections (or follow the PLOS ONE guidelines): abstract, introduction, methods (study population, theoretical framework, and statistical method), ethics, results, discussion, conclusion, and references.

Author response: Thank you for your valuable feedback. We appreciate your suggestions regarding the organization of the paper. We revised the manuscript to adhere to the recommended structure, including the sections you have outlined. This restructuring enhanced the clarity and coherence of the paper, ensuring that readers can more effectively navigate and comprehend our research findings. 

Concern # 2: The introduction section needs to end up with identifying the research gap, study objective(s), followed by the three research questions.

Author response: Thank you for your constructive feedback. We acknowledge the importance of clearly defining the research gap, study objectives, and research questions in the introduction section of our paper. In our revision, we ensured that the introduction concludes with a concise identification of the research gap, clearly stated study objective, and the three research questions. This provides a comprehensive understanding of the purpose and focus of our study right from the outset, facilitating better engagement with our research findings. We appreciate your guidance in refining the structure of our paper to enhance its clarity and effectiveness.

Author action:

The main shortcoming of the cited studies is that they restricted their analysis to a specific category of food products without considering the potential effects on the full shopping basket; that is, they assumed weak separability of the selected categories from the total food basket. These interdependencies are important because they help retailers to allocate their promotional budget across categories. Therefore, studying the full basket is a crucial component of the creation of any promotional mix in which retailers build new product bundles, provide special discounts, or put products in the best possible locations. To the best of our knowledge, very few studies have considered the effects of prices and promotions on the entire shopping basket. Drèze et al. (42) used panel data on household purchases from four stores of the same retailer, and Revoredo-Giha et al. (43) analysed the overall effect of promotions on consumers’ food purchases in Scotland by area of deprivation to formulate food and health policies. In this study, we propose a conceptual framework (Fig 2) of the effect of sales promotions and prices on household expenditures to achieve our research objective: to determine the effect of sales promotions on households’ allocation of their budget among the different products in a typical shopping basket in Spain. Our framework has two dimensions: (A) the direct effects of price and promotions on household expenditures, and (B) the possible cross-effects of price and promotions (i.e., substitution or complementarity), which could lead households to reallocate their budget.

Fig 2. Conceptual framework of this study.

Ultimately, we aim to contribute to the literature empirically and methodologically. Empirically, we aim to study household budget allocation decisions in Spain because retailers should be aware of how consumer responses to sales promotions vary between nations, states and provinces, as sales promotions are generally acknowledged as challenging to standardise due to the varying legal, economic and cultural factors that influence households and consumer behaviour. Consequently, we seek to provide retailers insights into the categories that are influenced most by promotions and how households manage their shopping budgets. Methodologically, we use the EASI demand system because it allows for flexibility of Engel curve shapes, unlike the AIDS demand model. Banks et al. (44) demonstrated, using nonparametric regressions, that Engel curves can have different shapes. Additionally, budget share error terms can be interpreted as unobserved preference heterogeneity, as permitted by the implicit Marshallian demand system developed by Lewbel et al. (45) that merges the proprieties of the Hicksian and Marshallian demand functions. In addition, we introduce promotional indices into the EASI demand system that allow for the estimation of the own-promotion and cross-promotion elasticities for a broader number of food categories. 

Therefore, we will strive to answer the following three major research questions:

 To what extent do promotion and price induce households to change their shopping expenditures? 

 Which product category has the strongest effect on expenditures when sold under a promotion?

 What are the possible cross-effects of promotions on the product categories in the shopping basket?

Concern # 3: The study population can be presented by a flowchart, showing number of households and/or household purchases, with exclusion criteria. Currently, it is not clear whether the authors included alcoholic beverages as part of household food purchases (I don’t think alcoholic beverages can be included as part of household food basket or purchases). What other food/beverage products need to be excluded (e.g., products purchased during Eister or Christmas), or weekly household purchases less than a certain amount of money or some other products that the authors excluded or may exclude them (e.g., foods for pets).

Author response: We would like to appreciate your comments, which contribute to the improvement of our article. For this question, we took into consideration the comment that you’ve raised, we added a flowchart showing the number of the households and the exclusion criteria to reach the final sample size.

We excluded household’s supplies (kitchenware, detergent, soap, insect repellent, and health and beauty care) since these categories represent a really small budget share and usually not purchased in a weekly basis which will provoke a big percentage of zeros in the dependent variable (expenditure share) of the EASI demand system.

In relation to the alcoholic beverages, they are included in the category “Drinks” as mentioned in the (S1 Appendix). As it’s a common classification in various papers; Drèze et al.(2004), and Revoredo-Giha et al.(2018) that included “Alcohol” as a category in their shopping basket analysis. And as indicated by Du and Kamakura (2008) that an in-depth analysis of the household budget allocation should include as complete a set of consumption categories as possible. 

Author Action: Flowchart of the household selection process.

Fig 2. Flowchart of the household selection process

(Check the response to reviewers file)

Concern # 4: A theoretical framework (or conceptual framework) can be useful to guide the conceptual relationships between promotions and price and the outcome of interest, in line with the study objective(s) and/or research questions.

Author response: We would like to appreciate your comments, we took into consideration the comment that you’ve raised, we added the conceptual framework of the study to show the objective and research questions of the study into the introduction.

Author Action:

Fig 3. Conceptual framework of the study

(Check the response to reviewers file)

Concern # 5: Also, the authors need to provide a definition for retail promotions (I guess promotion was self-reported by households in this study).

Author response: Thank you for your comments, the promotions are presented in the scanner panel database are self-reported; and presented as a categorical variable, that takes the value of 1 if the household’s made a transaction with a promotion and takes a value of 0, otherwise. This is a common case for all the scanner panel data as mentioned in Muth et al., (2019), that households in scanner panel data indicate whether they used some kind of promotion.

Author Action:

To identify and track products, a Universal Product Code (UPC) is employed, where each household were equipped with a barcode reader that allowed them to scan the UPCs and report all the information present in their shopping ticket; quantities, prices and a promotional variable that refers to retailers’ promotions, presented as a categorical variable that takes the value of 1 when the household purchase under promotion and 0 otherwise. The dataset also includes product characteristics (such as type of packaging, brand and flavour), and some socio-demographic characteristics of the household such as age, social class, presence of children, region, etc.

Concern # 6: Statistical methods: The outcome of interest, predictors, and statistical models (the expenditure regression, and EASI) need to be fully described. Some formulas, which are not in line with the research questions, can be moved to supplementary materials (e.g., the formula on weighted average price). In the regression model, variables on socioeconomic status and demographic of households need to be included, in addition to the variables on promotion and price, because currently the expenditure regression does not account for such important explanatory variables (for example, household food expenditures can be affected by household size).

Author Response: Thank you for the valuable comment, the formulas in relation to the data preparation (category price index, promotion index with category expenditures have been moved to the supplementary materials).

In relation to the regression model; the socioeconomic status and demographic of the households couldn’t be included in the regression as mentioned in the paper, that it is a fixed effect regression model. We added more explanation below the model to explain the fixed effect regression and the Hausman test that sustained why we choose the fixed effect regression over random effects.

Author Action: 

(Check the response to reviewers file)

Concern # 7: The results should present baseline characteristics of households as well as frequency of household food purchases (by food categories, prices, quantities, etc.), followed by results in response to each of the research questions in tables/figures that are well organized and clearly presented (the authors can have a look at the published papers for examples of tables and figures). The authors should use terms that are easily understood and are meaningful to a general audience of the paper (all readers are not econometricians to understand coefficients, significance shown by ***, or other technical jargons).

Author Response:

1. Thank you for the comments, they have been taken into consideration. We added average number of shopping trips, average price and quantity. We added a discussion of the added statistics too.

2. A note indicating the meaning of the stars is added in bottom of each table. Additionally, the standard errors are added in a new column.

Author Action:

(Check the response to reviewers file)

Concern # 8: Discussion: The first paragraph may cover an overall summary of the findings in the study, followed by comparison of the findings with relevant literature and possible explanations for discrepancies and similarities in the findings, with references. The study limitations and strengths, and the findings’ significance on public health nutrition policies can be discussed. Further research highlighted.

Author Response: Thank you for your thoughtful feedback. We appreciate the opportunity to address the comments regarding the discussion section of our study. In response to the reviewer's suggestion, we have revised the discussion section to provide a more comprehensive overview of our findings and their implications. Firstly, we have included an overall summary of the key findings of our study. Secondly, we have incorporated a comparison of our findings with existing literature. Additionally, we have discussed possible explanations for any disparities observed between our findings and those of previous studies. Furthermore, we have critically examined the limitations of our study. By acknowledging the constraints and biases inherent in our research design, we aim to provide a transparent assessment of the reliability and validity of our findings. 

Importantly, in relation to the discussion of the significance of our findings for public health nutrition policies. We tried to drive some conclusions, but the classification of the food categories from the beginning was not intended for studying the effect of promotion on the healthy eating and purchasing behavior. But we consider it as an interesting idea for future research.

Overall, we believe that this revision enhances the discussion section of our study, providing a more thorough analysis of our findings and their implications. We are grateful for the reviewer's insightful feedback, which has helped to strengthen our manuscript.

Author Response:

Discussion 

In this study, we present estimates of the impacts of both price and promotion on household expenditures for all items in the shopping basket, as well as on the allocation of the shopping budget across the shopping basket items. To do so, we used two complementary quantitative tools. First, we estimated a fixed-effect regression model to study the impact of price and promotion on household expenditures in our selected retailer outlets. Second, we used the censored EASI demand system established by Lewbel and Pendakur (45) to calculate the expenditure, price and promotion elasticities in order to analyse the cross-product effects of price and promotion, considering the censored nature of the dependent variable, which, in this study, was the expenditure share. 

The results of the fixed-effect regression indicate that both price and promotion had a positive effect on the household expenditures. The price elasticities were lower than unity, indicating the price-inelastic nature of the food products and that the households were not price-sensitive. This result further indicates that the food products were weakly separable from each other. This result is in line with that of Revoredo-Giha et al. (43) and in contrast with the result of Drèze et al. (42), which showed a negative effect of price on household expenditures and more price-sensitive consumers. These contrasting results are probably due to the different countries studied and the difference in the product aggregation, which, in (42) , was focused on the retailers’ product assortment, and in the current study, was based on the Spanish nutrition guidelines. 

As for promotion, various studies (41,52) reported its positive impact on household expenditures, as explained by the retailers’ promotional objective of enhancing their establishments’ sales in other words, pushing households to spend more while shopping. 

The results of the EASI demand system showed that demand for all the food categories, except for the ‘Residual category’, was inelastic in response to price changes, similar to (53,54) in Vietnam, (55) in the USA and (56) in Spain. ‘Drinks’ showed the highest own-price elasticity, as mentioned in (43) and (57). Contrary to the findings of (56,58) in Spain, ‘Milk, milk products and milk imitation products’ had the second highest elasticity, similar to (42,57,59,60). In any case, these comparisons are not straightforward, as the food aggregations differ across studies.

As for the cross-price elasticities in this study, many of them were quite low and insignificant. Moreover, the most significant cross-price elasticities revealed complementary relationships, as we expected, considering the broad number of food categories we considered. Substitution relationships were expected in each food category, while complementarity or even independent relationships were expected between categories, in line with (55). In any case, the cross-price elasticities were much smaller in absolute value relative to the own-price elasticities, as in Molina (61).

From the expenditure elasticities, we saw that all the food categories behaved as normal goods, which is also quite consistent with our expectations, as we are dealing with food categories and not with specific food products, among which we could expect some inferior goods behaviour.

---

## [Decision Letter · Decision Letter 1]

5 Apr 2024

PONE-D-23-33462R1Allocation of the household food budget among shopping basket items: How is it influenced by promotions?PLOS ONE

Dear Dr. Mehaba,

Thank you for submitting your manuscript to PLOS ONE. After careful consideration, we feel that it has merit but does not fully meet PLOS ONE’s publication criteria as it currently stands. Therefore, we invite you to submit a revised version of the manuscript that addresses the points raised during the review process.

I thank the authors for improving the paper and addressing reviewers’ comments. However, before further consideration, please address the following comments:Some in-text citations still do not conform to PLOS ONE’s style. Authors last name should no be written next to numbers.In table 2, please add frequency (N). Since you followed the two-step procedure suggested by Heien and Wessells (1990), please discus how it differs from Heckman sample selection model and address (Vermeulen, 2001) note in your model case.

Vermeulen, F. (2001). A note on Heckman-type corrections in models for zero expenditures. *Applied Economics*, *33*(9), 1089–1092. https://doi.org/10.1080/00036840010004004

Add the result of the probit model in the appendix sectionI could not find the coefficients value of the IMR and Z (sociodemographic) variables as stated in equation (4). Also, elaborate if you corrected the standard error.Add regression standard error in table 3.Please mention the method used in calculating elasticities' standard error.Please submit your revised manuscript by May 20 2024 11:59PM. If you will need more time than this to complete your revisions, please reply to this message or contact the journal office at plosone@plos.org. Please include the following items when submitting your revised manuscript:A rebuttal letter that responds to each point raised by the academic editor and reviewer(s). You should upload this letter as a separate file labeled 'Response to Reviewers'.A marked-up copy of your manuscript that highlights changes made to the original version. You should upload this as a separate file labeled 'Revised Manuscript with Track Changes'.An unmarked version of your revised paper without tracked changes. You should upload this as a separate file labeled 'Manuscript'.If applicable, we recommend that you deposit your laboratory protocols in protocols.io to enhance the reproducibility of your results. Protocols.io assigns your protocol its own identifier (DOI) so that it can be cited independently in the future. For instructions see: https://journals.plos.org/plosone/s/submission-guidelines#loc-laboratory-protocols. Additionally, PLOS ONE offers an option for publishing peer-reviewed Lab Protocol articles, which describe protocols hosted on protocols.io. Read more information on sharing protocols at https://plos.org/protocols?utm_medium=editorial-email&utm_source=authorletters&utm_campaign=protocols.

We look forward to receiving your revised manuscript.

Kind regards,

Mohammed Al-Mahish, Ph.D.

Academic Editor

PLOS ONE

Journal Requirements:

Reviewers' comments:

Reviewer's Responses to Questions

**Comments to the Author**

1. If the authors have adequately addressed your comments raised in a previous round of review and you feel that this manuscript is now acceptable for publication, you may indicate that here to bypass the “Comments to the Author” section, enter your conflict of interest statement in the “Confidential to Editor” section, and submit your "Accept" recommendation.

Reviewer #2: (No Response)

Reviewer #3: All comments have been addressed

2. Is the manuscript technically sound, and do the data support the conclusions?

Reviewer #2: (No Response)

Reviewer #3: (No Response)

3. Has the statistical analysis been performed appropriately and rigorously? 

Reviewer #2: (No Response)

Reviewer #3: (No Response)

4. Have the authors made all data underlying the findings in their manuscript fully available?

Reviewer #2: (No Response)

Reviewer #3: (No Response)

5. Is the manuscript presented in an intelligible fashion and written in standard English?

Reviewer #2: (No Response)

Reviewer #3: (No Response)

6. Review Comments to the Author

**Reviewer #2:** The authors did a great job considering the comments and suggestions from the reviewers. While my initial comment necessitated significant revision, I'm now inclined to accept the article for publication, given its substantial enhancement.

**Reviewer #3:** (No Response)

7. PLOS authors have the option to publish the peer review history of their article (what does this mean?). If published, this will include your full peer review and any attached files.

Reviewer #2: **Yes: **Sina Ahmadi Kaliji

Reviewer #3: No

---

## [Author Response · Author response to Decision Letter 1]

30 Apr 2024

Dear Editor, 

Thank you for allowing a resubmission of our manuscript, with an opportunity to address the comments. We would like to thank you for the great interest you have shown in our work, as well as the relevant remarks to improve the article.

Our point-by-point response to the comments are presented below:

Concern #1: Some in-text citations still do not conform to PLOS ONE’s style. Authors last name should not be written next to numbers.

Author response: Thank you for bringing that to my attention. I ensured that all in-text citations adhere to PLOS ONE’s style guidelines by removing authors' last names next to numbers. I appreciate your feedback.

Concern #2: In table 2, please add frequency (N).

Author response: Thank you for your feedback. We incorporated the frequency (N) into Table 2 as per your suggestion. This addition will provide readers with a clearer understanding of the sample size for each variable. 

Concern #3: Since you followed the two-step procedure suggested by Heien and Wessells (1990), please discus how it differs from Heckman sample selection model and address (Vermeulen, 2001) note in your model case.

Vermeulen, F. (2001). A note on Heckman-type corrections in models for zero expenditures. Applied Economics, 33(9), 1089–1092. https://doi.org/10.1080/00036840010004004

Author response: Thank you for your insightful comment. Indeed, the two-step procedure proposed by Heien and Wessells (1990) offers a distinct approach to address sample selection bias compared to the Heckman sample selection model. While both methods aim to mitigate the effects of sample selection bias, they differ in their underlying assumptions and estimation techniques. It has been shown that the Heckman two-step procedure only includes the participating households in the second step of the model. This may cause concerns with the efficiency of the method and the estimated results might only suit the households that participated. The Heien and Wessells two step model is based on the Heckman’s approach but it uses the same households in both of the steps, thus it solves the problem of the Heckman two-step model mentioned above. 

According to Heien and Wessells (1990), their approach improved results based on the goodness-of-fit and elasticity values. Even though Wessells' procedure may have inconsistent estimators according to Vermeulen (2001) but the Heckman-selection approach of Heien and Wessels (1990) is supported by Tauchmann (2005) Monte Carlo study of censored demand systems, where he indicates that the approach of Heien and Wessels (1990) has a good performance and might be one of the best choices for practical applications. Additionally, Fernando (2010) mentioned that Heien and Wessells (1990) is one of the two-steps procedures that minimizes computational time. Last but not least, the method of Heien and Wessels (1990) has been widely used by diverse recent works, to cite (Ni Mhurchu et al., 2013), (Sharma et al., 2014), (Caro et al., 2020) and (Gido, 2022).

We made sure to add a note in the manuscript in relation to the method: even though criticized by (51) but it has been supported by (52) indicating that is one of most suitable methods to account for censoring. Therefore we added the (Vermeulen, 2001) and (Tauchmann, 2005) references to reference list.

Concern #4: Add the result of the probit model in the appendix section

Author response: Thank you for your suggestion. I included the result of the probit model in the appendix section as per your recommendation. 

Concern #5: I could not find the coefficients value of the IMR and Z (sociodemographic) variables as stated in equation (4). Also, elaborate if you corrected the standard error.

Author response: Thank you for bringing this to our attention. We did not include these coefficients in the results section because we did not report the other coefficients of the equation (4), but we just included the elasticities that we computed form the coefficients of the EASI demand system (equation 4) which are stated in equation 9, 10 and11. That explains why the mentioned coefficients were not included. However, here are the coefficients for you to check them. 

 (Table in the response to reviewers file)

Concern #6: Add regression standard error in table 3.

Author response: Thank you for your comment. The regression standard errors have been included in Table 3 of the manuscript already. This query has been addressed within the manuscript. 

Concern #7: Please mention the method used in calculating elasticities' standard error.

Author response: Thank you for your comment. The standard errors for the elasticities were calculated based on bootstrapped standard errors with 500 replications. This method was employed to ensure robustness and reliability in our estimation. Additionally, we added the method under the tables.

References 

Caro, J.C., Valizadeh, P., Correa, A., Silva, A., Ng, S.W., 2020. Combined fiscal policies to promote healthier diets: Effects on purchases and consumer welfare. PLoS ONE 15. https://doi.org/10.1371/journal.pone.0226731

Fernando, J., 2010. University of Alberta Three Essays on Canadian Household Consumption of Food Away From Home Department of Rural Economy.

GIDO, E.O., 2022. Household Demand System of African Indigenous Vegetables in Kenya. Review of Agricultural and Applied Economics 25. https://doi.org/10.15414/raae.2022.25.01.94-103

Ni Mhurchu, C., Eyles, H., Schilling, C., Yang, Q., Kaye-Blake, W., Genç, M., Blakely, T., 2013. Food Prices and Consumer Demand: Differences across Income Levels and Ethnic Groups. PLoS ONE 8. https://doi.org/10.1371/journal.pone.0075934

SHARMA, A., HAUCK, K., HOLLINGSWORTH, B., SICILIANI, L., 2014. THE EFFECTS OF TAXING SUGAR‐SWEETENED BEVERAGES ACROSS DIFFERENT INCOME GROUPS. HEALTH ECONOMICS 23, 1159–1184.

Tauchmann, H., 2005. Efficiency of two-step estimators for censored systems of equations: Shonkwiler and Yen reconsidered. Applied Economics 37, 367–374. https://doi.org/10.1080/0003684042000306987

Vermeulen, F., 2001. A note on Heckman-type corrections in models for zero expenditures. Applied Economics 33, 1089–1092. https://doi.org/10.1080/00036840010004004

---

## [Editor Report · Decision Letter 2]

6 May 2024

PONE-D-23-33462R2Allocation of the household food budget among shopping basket items: How is it influenced by promotions?PLOS ONE

Dear Dr. Mehaba,

Thank you for submitting your manuscript to PLOS ONE. After careful consideration, we feel that it has merit but does not fully meet PLOS ONE’s publication criteria as it currently stands. Therefore, we invite you to submit a revised version of the manuscript that addresses the points raised during the review process.

 The coefficients of the IMR and other socioeconomics variables were not requested for me to check them as stated in your cover letter, but they were requested to make the results and the methodology of your study transparent and reproducible to other researchers.I noticed that in the probit model and the final model you changed the nature of the age variable from categorical variable as reported in table 2 to a numerical variable without informing the readers of such change, which obscures the reproducibility of the paper by other interested researchers. Also, the household size variable has 4 respondents who did not answer the question and the authors treated those who did not answer as a regular categorical level, which is incorrect. This is because this category level has few respondents and does not have enough statistical power to compare it and make inference with other categories. Also, the norm when dealing with respondents who do not answer questions is to keep the information as missing and not create a new categorical level. Thus, please re-estimate the model to address this issue.Report the results of residual standard error.

We look forward to receiving your revised manuscript.

Kind regards,

Mohammed Al-Mahish, Ph.D.

Academic Editor

PLOS ONE

---

## [Author Response · Author response to Decision Letter 2]

13 May 2024

Thank you for allowing a resubmission of our manuscript, with an opportunity to address the comments. We would like to thank you for the great interest you have shown in our work, as well as the relevant remarks to improve the article.

Our point-by-point response to the comments are presented below:

Concern #1: The coefficients of the IMR and other socioeconomics variables were not requested for me to check them as stated in your cover letter, but they were requested to make the results and the methodology of your study transparent and reproducible to other researchers.

Author response: Thank you for highlighting that. We've taken note of your feedback, and we've now included the coefficients of the IMR and other socioeconomic variables in the document for transparency and reproducibility as requested. 

(Table in the response to reviewers file)

Concern #2: 

1. I noticed that in the probit model and the final model you changed the nature of the age variable from categorical variable as reported in table 2 to a numerical variable without informing the readers of such change, which obscures the reproducibility of the paper by other interested researchers.

2. The household size variable has 4 respondents who did not answer the question and the authors treated those who did not answer as a regular categorical level, which is incorrect. This is because this category level has few respondents and does not have enough statistical power to compare it and make inference with other categories. Also, the norm when dealing with respondents who do not answer questions is to keep the information as missing and not create a new categorical level. Thus, please re-estimate the model to address this issue.

Author response: 

1. Thank you for bringing this to our attention. You're correct that the age variable is categorical (as reported in Table 2) and continuous in the probit model and the final model which may create confusion for the readers. But the age variable present in our dataset is continuous and the age groups created in Table 2 were used just as descriptive of the sample. In any case we removed the age groups created and add the mean of age in Table 2 to avoid any confusion. We apologize for any confusion caused by not explicitly mentioning it in the paper. 

2. Thank you for bringing this to our attention. You're absolutely right, and we apologize for the oversight. We acknowledge the importance of handling missing data appropriately and understand that treating non-responses as a separate categorical level may not be statistically sound. 

Accordingly, we used the information from previous years of the dataset that we have knowing that households stay in the panel for a period up to three years we checked the households’ size for the four missing households and we found out that they have between more than 5 and 6 members so we added them to the +5 category ( As can be seen in the pictures attached, the one in the right is the dataset that we used and the one in the left is the data we used to replace the missing in our current dataset). We re-estimated the model, taking into account your feedback. The results are presented in the manuscript and here. The difference between the estimations is very tiny and small as you can see in the pictures attached below. We thank you for your constructive criticism, which will undoubtedly improve the quality of our research.

(Pictures attached the response to reviewers file)

Concern #3: Report the results of residual standard error.

Author response: Thank you for your keen observation. The residual standard error has been included in the Table S3 in the appendix as requested.

---

## [Editor Report · Decision Letter 3]

21 May 2024

Allocation of the household food budget among shopping basket items: How is it influenced by promotions?

PONE-D-23-33462R3

Dear Ms. Mehaba,

We’re pleased to inform you that your manuscript has been judged scientifically suitable for publication and will be formally accepted for publication once it meets all outstanding technical requirements.

Kind regards,

Mohammed Al-Mahish, Ph.D.

Academic Editor

PLOS ONE
---

## [Editor Report · Acceptance letter]

27 May 2024

PONE-D-23-33462R3 

PLOS ONE

Dear Dr. Mehaba, 

I'm pleased to inform you that your manuscript has been deemed suitable for publication in PLOS ONE. Congratulations! Your manuscript is now being handed over to our production team.

Kind regards, 

on behalf of

Dr. Mohammed Al-Mahish 

Academic Editor

PLOS ONE